# Allele-Specific PCR for Detection of Missense Mutations in the Chimeric *BCR::ABL1* Gene Causing Failure of Tyrosine Kinase Inhibitor Therapy in CML Patients

**DOI:** 10.3390/ijms26083728

**Published:** 2025-04-15

**Authors:** Anastasia Skripkina, Irina Fevraleva, Elena Kuzmina, Bella Biderman, Elena Stepanova, Ekaterina Chelysheva, Anna Turkina, Andrey Sudarikov

**Affiliations:** National Medical Research Center for Hematology, 125167 Moscow, Russia; skripkina.a@blood.ru (A.S.);

**Keywords:** chronic myeloid leukemia, AS-PCR, transcript *BCR::ABL1*, mutations

## Abstract

Missense mutations in the *BCR::ABL1* kinase domain are found in approximately 12–80% of patients with chronic myeloid leukemia (CML). Clinically significant mutations include T315I, M244V, Y253H/F, E255K/V, V299L, and F359V. The aim of this study was to create a diagnostic system for rapid and inexpensive detection of the above mutations. We used genomic DNA and RNA from peripheral blood and bone marrow cells of 57 patients with a Ph-positive CML diagnosis established in the chronic phase. We have developed a method to detect mutations in the *BCR::ABL1* gene based on allele-specific real-time polymerase chain reaction (AS-PCR). In parallel, we analyzed the RNA sequence of the protein kinase domain of the same samples by next-generation sequencing (NGS) covering the points of putative mutations. In this work, we compared the results obtained by both methods for mutation detection and variant allele frequency (VAF) estimation of mutated vs. normal alleles. The sensitivity and specificity of our diagnostic system were also evaluated. It was found that AS-PCR gives reliable results at VAF up to 0.01%. AS-PCR has high sensitivity and may serve as an alternative for the more time-consuming NGS in some cases, as well as for monitoring CML treatment and for analyzing archival material.

## 1. Introduction

Missense mutations in the *BCR::ABL1* kinase domain associated with resistance to tyrosine kinase inhibitor (TKI) therapy are found in approximately 12–35% of patients with therapy failure in the chronic phase and in 50–80% of patients in the acceleration and blast crisis phases [1,2]. Such a wide range of reported mutation frequencies is associated not only with the stage of disease and individual patient characteristics but also with the sensitivity and specificity of diagnostic methods [3,4,5].

The majority of mutations responsible for the failure of TKI therapy affect nine amino acids of the protein kinase domain encoded by the chimeric *BCR::ABL1* gene [6]. One of the most common and clinically significant mutations is the pan-resistant T315I mutation, which causes resistance to both 1st generation TKI (imatinib) and 2nd generation TKIs (nilotinib, dasatinib, and bosutinib) [7,8]. According to the literature, the detection frequency of the T315I mutation varies from 2% to 20% [9]. In addition to T315I, the most frequent mutations are M244V, Y253H/F, E255K/V, V299L, and F359V, conferring different sensitivity to certain TKIs [10,11,12,13].

In the presence of combined mutations, the tumor clone becomes more resistant to the effects of 1st and 2nd generation TKIs. In such cases, 3rd generation TKIs (ponatinib) and the allosteric TKI asciminib may be effective [14].

Thus, early detection of mutations leading to the emergence of TKI resistance provides physicians with a tool for selecting individualized therapy.

Currently, next-generation sequencing (NGS) is the primary method for detecting 47 missense mutations [15]. Despite its versatility, this technique can be time-consuming and costly. To address the issue of detecting clinically significant mutations, we propose the use of sensitive, fast, and inexpensive allele-specific polymerase chain reaction (AS-PCR) [15,16,17]. Thus, the goal of this study was to develop a diagnostic system for detecting clinically significant mutations in the chimeric *BCR::ABL1* gene, such as T315I, M244V, Y253H, V299L, and F359V, as an alternative to NGS in certain cases. The system is based on AS-PCR and can use both cDNA and genomic DNA, facilitating the analysis of archived materials.

## 2. Results

We developed a system of primers for the rapid detection of the most frequent clinically significant mutations of the chimeric *BCR::ABL1* gene conferring resistance to TKI therapy (Table 1). Primers for quantitative TaqMan PCR were selected by analyzing the *ABL1* gene sequence in the vicinity of mutation points [18]. The last nucleotide at the 3′ end of each primer corresponded to a normal or mutant nucleotide in the *ABL1* gene sequence. The penultimate nucleotide of the 3′ end of each primer was modified for greater specificity of the AS-PCR.

The sensitivity of the proposed system was evaluated by serial dilutions of the patient’s mutated DNA with the DNA of a healthy donor (Figure 1 and Figure 2). Figure 1 shows the amplification curves in the experiment with tenfold dilution from the first point using the DNA of a patient with the T315I mutation. The difference in threshold cycles between the amplification curves of the healthy donor DNA obtained in AS-PCR with primers for normal alleles (green curves in Figure 1, marked by arrow 1) and primers for detection of mutant alleles (red curves in Figure 1, marked by arrow 1) is equal to 15 cycles. Threshold cycles of DNA amplification curves with primer for mutation detection at a 10^5^-fold dilution of patient DNA (blue curves in Figure 1, marked by dotted arrow 2.5) coincide with the threshold cycle of the corresponding curve of healthy donor DNA (red curves in Figure 1, marked by arrow 1). Thus, at an allele ratio of 0.003% or less, we may report the absence of mutant alleles. The threshold cycles of DNA amplification curves corresponding to a 10^4^-fold dilution of the DNA of the patient with the mutation (blue curves in Figure 1, marked by the dotted arrow 2.4) reproducibly indicate the presence of the mutation. Consequently, at an allele ratio greater than 0.01%, we can confidently assert the presence of a mutation. Such sensitivity of T315I mutation detection is in full agreement with what is required for clinical diagnosis [19]. Similar results were obtained for other mutations. 

According to recent guidelines, NGS performed on cDNA is considered to be the gold standard for detecting mutations that confer resistance to TKI [15]. In our work we compared mutation data obtained on DNA from peripheral blood or bone marrow using the AS-PCR method and the NGS DNA and cDNA mutation data (Table 2).

In the case of AS-PCR, the results for VAF estimation were close to those obtained by NGS in the cases with VAF > 5%. VAFs lower than 5% were NGS undetectable. AS-PCR, on the other hand, gave reliable results for VAF up to 0.01%.

## 3. Discussion

We compared the VAF values of mutations obtained using AS-PCR versus NGS after either nested or single-stage PCR on DNA or cDNA. The VAF calculated from NGS data was significantly higher than the VAF measured by AS-PCR. This result is expected and can be explained by the fact that, in the case of AS-PCR, the VAF represents the ratio of mutated alleles to total alleles of the *ABL1* gene in a sample. Whereas in the case of cDNA analysis, VAF refers to the ratio of mutant and normal alleles in *BCR::ABL1* chimeric transcripts only. The VAF values obtained from genomic DNA analysis using AS-PCR and NGS methods (Table 2) were similar. However, the sensitivity of NGS in our analysis was 5%, while AS-PCR allowed for the determination of VAFs up to 0.01%.

Thus, the sensitivity of the AS-PCR method, which is more than 10 times higher than the sensitivity of sequencing, allows obtaining reliable results in real time about the presence of mutations at a VAF of up to 0.01%. Similar results are obtained by the NGS cDNA method; however, it is more expensive, labor-intensive, and involves multiple steps—RNA isolation, reverse transcription, subsequent nested PCR, NGS, and subsequent computer processing of the results.

The AS-PCR method is also indispensable for analyzing archival material when the RNA is degraded. In addition, the high sensitivity of the method allows monitoring of mutation during TKI therapy of patients or when changing the drug or detecting a new mutant clone that has emerged [20].

Below we present examples of allele ratio monitoring for different mutations in the chimeric gene of three CML patients with TKI-1 treatment failure (Table 3, Table 4 and Table 5). The row with the maximum allele ratio and *BCR::ABL1* transcript level is highlighted in bold.

The results of AS-PCR were confirmed by NGS; besides, it was performed less frequently.

## 4. Materials and Methods

### 4.1. Patients

DNA was isolated from peripheral blood and bone marrow cells of 57 patients treated at the National Medical Research Center for Hematology (Moscow, Russia) from 2022 to 2024. All patients had a diagnosis of Ph-positive CML and failure of 1st generation TKI therapy. *BCR::ABL1* chimeric transcript levels were at least 1% in all patients. Informed consents from patients to participate in the study were obtained. The study was conducted in accordance with the Declaration of Helsinki and with the approval of the Institutional Ethics Committee (protocol code 185 of 27 February 2025).

Peripheral blood mononuclear cells were obtained after lysis of erythrocytes from 2 to 8 mL of peripheral blood or bone marrow [21]. DNA and RNA isolation from leukocytes was performed using the reagent kit (Interlabservice, Moscow, Russia). Reverse transcription of isolated RNA to obtain cDNA was performed using a reagent kit (Interlabservice, Russia).

Quantitation of *BCR::ABL1* transcript level was performed using the Rotor-Gene amplifier (QIAGEN, Hilden, Germany) and the AmpliSense^®^ Leukemia Quant M-bcr-FRT reagent kit (Interlabservice, Russia).

### 4.2. Next-Generation Sequencing

Missense mutations were measured in the *BCR::ABL1* chimeric transcript and in the *ABL1* gene (in cDNA and DNA, respectively). The target *BCR::ABL1* region was amplified using either one-step PCR of sample DNA or “nested” PCR as described by van Dongen J. [22]. A panel of primers was created for the amplification of exons 4–9 of the *ABL1* gene (Table 6). The conditions for PCR were as follows: preheating—94 °C 300 s; subsequent thermocycling—35 cycles: denaturation 94 °C 30 s, annealing 60 °C 30 s, elongation 72 °C 90 s; final elongation 72 °C 5 min. Quality control of amplification products in both variants was performed by electrophoresis in 2% agarose gel. The amplicons obtained were used to create sequencing libraries using the Nextera XT DNA Library Preparation Kit and Nextera XT Index v2 (Illumina, San Diego, CA, USA). Sequencing was performed on a MiSeq genetic analyzer (Illumina, USA), and bioinformatic analysis was performed using the open-source software Trimmomatic (version 0.39) [23], BWA (version 0.7.17-r1188) [24], SAMtools (version 1.10) [25], Vardict (version 1.8.2) [26], and Annovar (release date 2020-06-08) [27]. Sequencing results were compared with the reference sequence NM_005157. The probable pathogenicity of detected mutations was analyzed using the Franklin by Genoox online database [28]. In the case of cDNA sequencing of the chimeric *BCR::ABL1* gene, the ratio of the number of reads with the mutation to the total number of reads determined the proportion of the mutated transcript. In the case of *ABL1* gene DNA sequencing, the ratio of the number of reads with a mutation to the total number of reads determined the variant allele frequency (VAF).

### 4.3. AS-PCR

All primers and fluorescent TaqMan probes were synthesized by «Syntol™» (Moscow, Russia). The sequences are summarized in Table 1. To detect each target mutation, two almost identical PCR reactions were performed, differing only by one primer for amplification of the normal or mutant allele. The conditions for AS-PCR were as follows: preheating—94 °C 300 s; subsequent thermocycling—45 cycles: denaturation at 94 °C 20 s, annealing and elongation at 65 °C 50 s. The amount of each primer—10 pmol per reaction, each probe—5 pmol; reaction volume—25 μL. PCR buffer, MgCl_2_ dNTP, and Taq polymerase (Syntol, Moscow, Russia) were used according to the manufacturer’s instructions. AS-PCR was performed on a CFX96 C1000 Touch instrument (Bio-Rad Laboratories, Hercules, CA, USA).

### 4.4. Allele Ratio and VAF in AS-PCR Assays

Allele ratio (AR)—the ratio of the number of alleles carrying a mutation in the *ABL1* gene to the number of *ABL1* gene alleles without mutations was calculated using the formula: AR = (2^C*ABL1*^:2^C*ABL1*mt^) × 100%, where C*_ABL1_* is the PCR threshold cycle for the normal gene and C*_ABL1_*_mt_ is the threshold cycle for the *ABL1* gene with a certain mutation.

ΔCT was the difference between the threshold cycle (Ct) value for the mutant allele and the Ct value for the normal allele.

Variant allele frequency (VAF)—i.e., the ratio of the proportion of alleles carrying the mutation in the *ABL1* gene to all alleles of the *ABL1* gene both with and without the mutation—was calculated using the formula: VAF = 100% × AR: (100% + AR), where AR is the allele ratio.

## 5. Conclusions

The developed diagnostic system based on allele-specific polymerase chain reaction with primers for detection of T315I, M244V, Y253H, V299L, and F359V mutations in the chimeric *BCR::ABL1* is a sensitive, specific, and simple method. The sensitivity of AS-PCR allows the detection of leukemia clones with mutation levels up to 0.01%. This analysis is useful prior to choosing farther-targeted TKI drugs. The detection of low-percentage mutant tumor clones is also of great value, particularly when, against the background of TKI therapy, there is a selection of tumor clones with a mutation that is resistant to this therapy. This is especially relevant in the case of the panresistant T315I mutation, which can sometimes be detected even at the onset of CML in certain patients. Moreover, the abundance of this leukemia clone can increase or decrease, which is impossible to predict without continuous monitoring of the presence of mutations. It should also be noted that we are not proposing to replace sequencing with AS-PCR. In all PCR negative cases, samples will still be sent for NGS. However, in the majority of cases, critical information for making clinical decisions will be available faster and at a lower cost.

AS-PCR has another significant advantage over NGS: faster results, lower costs, and a simple research protocol. The AS-PCR method for diagnosing *BCR::ABL1* mutations can be applied in any laboratory worldwide, as the reaction requires only allele-specific primers and probes, as well as the creation of optimal PCR conditions. We propose to use our diagnostic system for primary screening of resistant CML cases and for monitoring TKI treatment of patients. This system allows us to work with both DNA and RNA. It also provides the opportunity to work with archival material when RNA degrades.

## Figures and Tables

**Figure 1 ijms-26-03728-f001:**
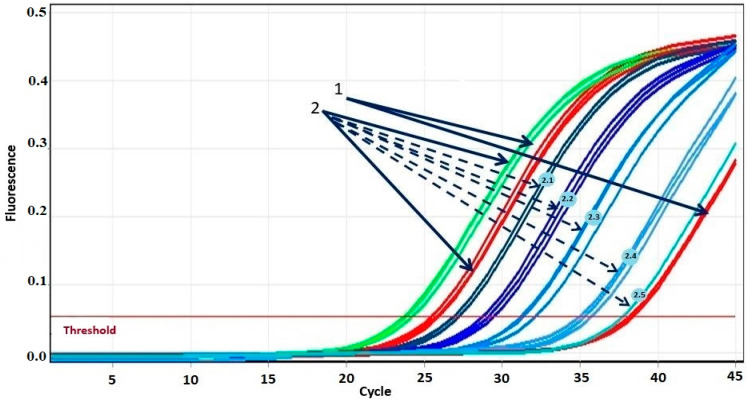
Amplification curves of real-time PCR of DNA from a patient with T315I mutation in different dilutions. 1—DNA amplification curves of a healthy donor during real-time PCR. 2—DNA amplification curves of a patient with the T315I mutation during real-time PCR. Solid arrows indicate DNA amplification curves without dilution with normal (green) and mutant (red) primers. The color gradient from dark blue to light blue indicates five amplification curves of consecutive DNA dilutions during real-time PCR with mutant primers: dashed arrow 2.1 indicates the curve corresponding to dilution 1/3; 2.2—dilution 1/10; 2.3—dilution 1/100; 2.4—dilution 1/1000; 2.5—dilution 1/10,000.

**Figure 2 ijms-26-03728-f002:**
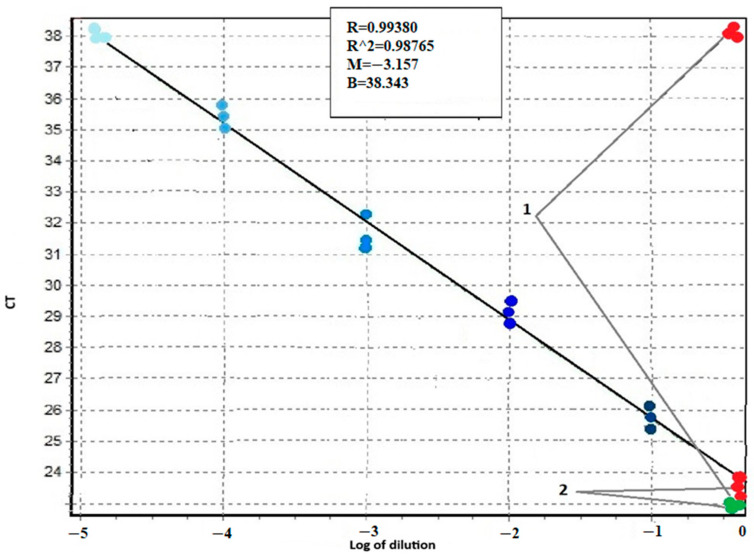
Calibration curve for determining the sensitivity of AS-PCR with primers detecting the T315I mutation of the *BCR::ABL1* chimeric gene. The X-axis is the decimal logarithm of the sample DNA dilution. On the Y-axis is the value of the CT threshold cycle. 1—threshold cycles for AS-PCR DNA of a healthy donor 2—threshold cycles for AS-PCR DNA of a patient with T315I mutation (green dots refer to normal alleles, red—to mutant alleles). The color gradient from dark blue to light blue indicates the points corresponding to the threshold cycles of PCR amplification with mutant primers for consecutive tenfold dilutions of DNA of the patient with the T315I mutation in the DNA of a healthy donor.

**Table 1 ijms-26-03728-t001:** Primers and fluorescent probes used for AS-PCR detection of mutations in the chimeric *BCR::ABL1* gene.

Amino Acid Substitution/Nucleotide Mutation/NCBI SNP	Primer/Probe Orientation	Sequence (5′ to 3′)
NM_005157.6 (*ABL1*):c.730A>G (p.Met244Val) rs121913456	Reverse w Reverse mt Forward Probe	5′-CGCCCAGCTTGTGCTTCTT-3′ 5′-CGCCCAGCTTGTGCTTCTC-3′ 5′-CTATGGTGTGTCCCCCAACT-3 5′-Cy5-GTCCGTGCG(T-BHQ2)TCCATCTCCCACTTG-(P)-3′
NM_005157.6 (*ABL1*):c.757T>C (p.Tyr253His) rs121913461	Reverse w Reverse mt Forward Probe	5′-CTCGTACACCTCCCCGAA-3′ 5′-CTCGTACACCTCCCCGAG-3′ 5′-CTATGGTGTGTCCCCCAACT-3 5′-Cy5-GTCCGTGCG(T-BHQ2)TCCATCTCCCACTTG-(P)-3′
NM_005157.6 (*ABL1*): c.895G>C (p.Val299Leu) rs1057519771	Reverse w Reverse mt Forward Probe	5′-TACTCACCAAGGAGCTGCTC-3′ 5′-TACTCACCAAGGAGCTGCTG-3′ 5′-TCCTTCTGCAGGAGGACACCATGGA-3′ 5′-Cy5-GGTTAGGGTG(T-BHQ2)TTGATCTCTTTCATGACTGC(P)-3′
NM_005157.6 (*ABL1*):c.944C>T (p.Thr315Ile) rs121913459	Reverse w Reverse mt Forward Probe	5′-TCCCGTAGGTCATGAACTCTG-3′ 5′-TCCCGTAGGTCATGAACTCTA-3′ 5′-GACAGTTGTTTGTTCAGTTGGGA-3′ 5′-Cy5-CAACAAGACAACGAGGACTTCAACACGTG-RTQ2-3′
NM_005157.6 (*ABL1*):c.1075T>G (p.Phe359Val) rs121913452	Forward w Forward mt Reverse Probe	5′-GGAGTACCTGGAGAAGAAAAAGT-3′ 5′-GGAGTACCTGGAGAAGAAAACGV-3′ 5′-CCTGAGACCTCCTAGGCTG-3′ 5′-Cy5-CAGCCTGCGCCATGGAGTCACAG-BHQ2-3′

**Table 2 ijms-26-03728-t002:** Comparison of VAF obtained by AS-PCR and NGS mutation studies.

Patient No.	VAF for DNA AS-PCR Amplification	VAF for NGS by One-Step DNA PCR	VAF for NGS by Nested cDNA PCR
Patient 1	23.8%—T315I	36.5%—T315I	96%—T315I
Patient 2	0.03%—M244V	Less than 5%—M244V	Less than 5%—M244V
Patient 3	0.32%—F359V	Less than 5%—F359V	Less than 5%—F359V
Patient 4	6.5%—T315I 26.7%—V299L	Less than 5%—T315I Less than 5%—V299L	Less than 5%—T315I; 17.5%—V299L
Patient 5	31.2%—T315I	20%—T315I	98%—T315I

**Table 3 ijms-26-03728-t003:** Allele ratio monitoring of M244V and F359V *BCR::ABL1* gene mutations for patient A.

Date of Analysis	ΔCT for the M244V Mutation	AR % for the M244V Mutation	ΔCT for the F359V Mutation	AR % for the F359V Mutation	*BCR::ABL1* Transcript %
6 June 2022	12.1	0.02	12.7	0.02	66.8
**10 August 2023**	**0.8**	**59.5**	**5.1**	**2.91**	**54**
25 September 2023	9.5	0.11	11.8	0.03	0.09
29 September 2023	11.6	0.03	12.3	0.02	0.07
29 November 2023	Hematopoietic stem cell transplantation				
27 December 2023	11.6	0.03	12.4	0.02	n/a
29 January2024	2.3	20.6	11.9	0.03	49.1
3 April 2024	11.9	0.03	12.7	0.01	n/a
11 July 2024	11.4	0.04	12.5	0.02	n/a

Patient A. Diagnosis: Resistant course of CML with an extramedullary lesion. M244V and F359V mutations were detected. Progression to lymphoid blast crisis on the background of nilotinib therapy in the 2nd line, relapse after hematopoietic stem cell transplantation. n/a—This sample has not been studied.

**Table 4 ijms-26-03728-t004:** Allele ratio monitoring of T315I *BCR::ABL1* gene mutation for patient N.

Date of Analysis	ΔCT for the T315I Mutation	AR % for the T315I Mutation T315I	*BCR::ABL1* Transcript %
**8 December 2022**	**4.3**	**5.2**	**55.549**
10 April 2023	3.6	8.3	35.27
20 December 2023	10.4	0.07	12.787
14 March 2024	11.2	0.04	3.143
6 June 2024	15.4	0	2.186
16 August 2024	14.7	0	1.462
15 November 2024	20	0	0.755

Patient N. Diagnosis: CML in chronic phase. Failure of therapy with three lines of TKI (imatinib, nilotinib, dasatinib); T315I mutation was detected. A subsequent change in therapy to a new TKI drug active against clones with T315I mutation resulted in a two-order-of-magnitude decrease in *BCR::ABL1* transcript value, and the allele ratio dropped to an undetectable value.

**Table 5 ijms-26-03728-t005:** Allele ratio monitoring of T315I *BCR::ABL1* gene mutation for patient B.

Date Analyzed	ΔCT for the T315I Mutation	AR % for the T315I Mutation T315I	*BCR::ABL1* Transcript %
**16 December 2022**	**3.5**	**9**	**92.796**
22 March 2023	2	24.8	56.4
21 June 2023	5	3.1	76.7
18 December 2023	3	12.4	63.5
11 April 2024	1.7	31.2	54.7
16 July 2024	4.8	3.7	46
12 November 2024	5.2	2.7	57.4

Patient B. Diagnosis: CML in chronic phase. Failure of therapy with four lines of TKI, T315I mutation was detected. A change in therapy to asciminib did not lead to the desired result.

**Table 6 ijms-26-03728-t006:** Primers used for PCR amplification of exons 4–9 of the *ABL1* gene.

Primers		Nucleotide Sequence of Primer (5′-3′)	Product Size, bp
1	abl1_ex4_FW	TGTGTAGTGAATTAAGGCTCAGC	456
1	abl1_ex4_RV	GAGGTAGACTTCCAGGCAGA	
2	abl1_ex5_FW	TCAGCTGTCATGGAACCTGT	350
2	abl1_ex5_RV	CCAACGAGGTTTTGTGCAGT	
3	abl1_ex6_FW	GGAGCAGAGTCAGAATCCTTC	399
3	abl1_ex6_RV	TGCCAGCACTGAGGTTAGAA	
4	abl1_ex7_FW	CTCAGCAGTGGTGGATTTGT	333
4	abl1_ex7_RV	GGAAGAGCAAGAAAGAGGCA	
5	abl1_ex8_FW	AGCCTTGTCCTGGTCTTCTG	396
5	abl1_ex8_RV	TGTACACACTCCTGCACAGT	
6	abl1_ex9_FW	CGTTTTGACTTGTTGCAGCA	375
6	abl1_ex9_RV	AATACTCCACACCTCTGCCC	

## Data Availability

Data is contained within the article.

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
