# Peer review of "Allele-Specific PCR for Detection of Missense Mutations in the Chimeric *BCR::ABL1* Gene Causing Failure of Tyrosine Kinase Inhibitor Therapy in CML Patients"

_ijms, 2025, doi:10.3390/ijms26083728_

Round 1
Reviewer 1 Report
Comments and Suggestions for Authors
The authors describe the development of an allele-specific PCR assay for the detection of resistance mutations in the BCR::ABL1 fusion gene. The method is well described and logically presented. The manuscript should be considered for publication in IJMS.
Comments / Questions for Clarification:
Throughout the manuscript, the terms "AS-PCR" and "AC-PCR" are used. Do the authors consider these terms synonymous, or do they refer to different methodological approaches? Please clarify the distinction, if any.
On page 2, line 47, you state that NGS is the primary method for the detection of missense mutations. However, on page 4, line 100, you refer to Sanger sequencing or NGS of cDNA as the gold standard.
On page 3, line 76, the determination of LOB and LOD was limited to a minimum, which is acceptable in this context, as the assay is primarily intended for qualitative rather than quantitative detection. Moreover, the reported values of 0.003% and 0.1% appear plausible and reasonable.
In Table 2, could the authors comment on why the VAF of both mutations detected by nested PCR in the patient 4 was not higher? Based on the use of nested PCR on cDNA, I would have expected higher VAF values compared to methods based on genomic DNA. Additionally, why was the 26.7% mutation not detected by NGS in patient 4? Only in patient 5 is the VAF detected by the cDNA-based method higher than that of the gDNA-based methods, as would be expected. I am aware that ratios based on cDNA and gDNA do not always follow the expected pattern, but could you please comment on these findings?
For my understanding and for the discussion: do you require five standard curves for five separate qPCRs in this assay, and a positive and a negative sample for each PCRs and all tests in triplicates? Is this approach still faster and more cost-effective compared to the nested PCR followed by Nextera library prep?
Reviewer 2 Report
Comments and Suggestions for Authors
The work presented by authors is inovative and suitable for publication in International Journal of Molecular Sciences. Before publication please consider my minor reccommandations:
- Correct the AC-PCR- several times the authors used this abbreviation
- At the first appearance in the text use the complete recommended nomenclature to describe each variant
- use one uniform description for the investigated variants- currently the authors used the term clinically relevant mutation/ clinically significant/ etc terms- most appropriate may be clinically actionable variants (for example)
- for reference no 18, 21, 28 use the complete description. For example for Ensembl recommendation- If you've used Ensembl in your work, please cite the most recent overview article and the Ensembl release you retrieved your data from (currently 113)- PMID: 39656687 being the most recent, please see the site recommendations for Ensembl and for the other two.
